Performance of multigene testing in cytologically indeterminate thyroid nodules and molecular risk stratification

Zhou Yuanyuan 1
Wu Xinping 2
Zhang Yuzhi 2
Li Zhiqiang 1
Ge Xia 3
Chen Hao 1
Mao Yuan 1
Ding Wenbo dinggnib@sina.com 2
1 Genome Center, KingMed Center for Clinical Laboratory Co., Ltd , Hefei , Anhui Province , China
2 Department of Ultrasound, Affiliated Hospital of Integrated Traditional Chinese and Western Medicine, Third Clinical Medical College, Nanjing University of Chinese Medicine , Nanjing , Jiangsu Province , China
3 Department of Pathological Diagnosis, KingMed Center for Clinical Laboratory Co., Ltd , Hefei , Anhui Province , China
Menini Stefano
Electronic publication date: 2023 Sep 18
Publication date: 2023
Volume: 11
Electronic Location ID: e16054
Received 2023 Mar 20; Accepted 2023 Aug 16
Copyright: ©2023 Zhou et al.
Copyright year: 2023
Copyright holder: Zhou et al.
License: This is an open access article distributed under the terms of the Creative Commons Attribution License, which permits unrestricted use, distribution, reproduction and adaptation in any medium and for any purpose provided that it is properly attributed. For attribution, the original author(s), title, publication source (PeerJ) and either DOI or URL of the article must be cited.
License URL: https://creativecommons.org/licenses/by/4.0/

Keywords: Indeterminate cytology, Molecular testing, Molecular risk stratification, Next-generation sequencing, Thyroid nodule

Funding: Medical Scientific Research Foundation of Jiangsu Province of China M2020102 Jiangsu Provincial Key Research and Development Program BE2020726 Thyroid Research Project of Young and Middle-Aged Doctors of China International Medical Exchange Foundation BQE-JZX-202115 This project was funded by the Medical Scientific Research Foundation of Jiangsu Province of China (Grant No. M2020102), the Jiangsu Provincial Key Research and Development Program (Grant No. BE2020726) and the Thyroid Research Project of Young and Middle-Aged Doctors of China International Medical Exchange Foundation (Grant No. BQE-JZX-202115). The funders had no role in study design, data collection and analysis, decision to publish, or preparation of the manuscript.

==============================
Objective

Thyroid cancer is the third most prevalent cancer among females. Genetic testing based on next-generation sequencing may provide an auxiliary diagnosis to reduce cytologically diagnostic uncertainty. However, commercial multigene tests are not widely available and are not well-tested in the Chinese population.

Methods

In this study, we designed a multigene testing panel and evaluated its performance in 529 cytologically indeterminate thyroid nodules (Bethesda III, IV and V). The molecular data of the DNA mutations and RNA fusions of fine needle aspiration samples were reviewed in conjunction with a clinical diagnosis, pathological reports, and definitive surgery for retrospective analysis. Then, the molecular risk stratification was investigated for its accuracy in malignant risk prediction.

Results

The overall combined consistency revealed substantial agreement (Kappa = 0.726) with the sensitivity, specificity, positive predictive value, and negative predictive values of 97.80%, 82.14%, 98.99%, and 67.65%, respectively. The most common aberration was BRAFV600E (82.59%), followed by NRAS mutants (4.07%), RET fusions (3.70%), and KRAS mutants (3.15%). Two cases (0.44%) were categorized into a high-risk group, 426 cases (94.67%) were categorized into a BRAF-like group with totally histopathologic papillary patterned tumors, and 22 cases (4.89%) were categorized into a RAS-like group with 14 papillary and eight follicular patterned tumors when the cohort concurrent aberrations were excluded. Potentially aggressive features may be related to concurrent molecular alterations of BRAFV600E with TERTQ302R, and AKT1L52R, NRASG12C, NRASQ61R, and CCDC6-RET fusions.

Conclusions

This study provided a multigene panel for identifying benign nodules from cytologically indeterminate thyroid nodules to avoid unnecessary surgery. We provide further evidence for using molecular risk stratification as a promising predictor of disease outcomes. The results of this study may be limited by the extremely high prevalence of cancer in the cohort for clinical reference.

Introduction

Thyroid cancer was the third highest prevalent cancer among females in 2022. Its incidence has increased dramatically but its mortality rate has remained low in both the United States and China since 2000 (Miller et al., 2022; Xia et al., 2022). As socioeconomic levels and radiologic technology have improved, the incidence rate of thyroid cancer has increased in transitioned countries over that in transitioning countries. This increase may be attributable to overdiagnosis (e.g., ionizing radiation), obesity, and exposure to hormones and environmental pollutants, etc. (Sung et al., 2021). The active surveillance for microcarcinoma (e.g., optimizing molecular markers and improved risk stratification) is recommended in place of frequent imaging screenings (Sung et al., 2021) due to the indolent property of small thyroid tumors.

Fine needle aspiration cytology (FNAC) is the gold standard technique for thyroid nodules with suspicious ultrasound features. However, its accuracy is limited in cytologically indeterminate samples (Ulisse et al., 2021). According to the latest version of The Bethesda System for Reporting Thyroid Cytopathology (TBSRTC), 20%∼30% of nodules may be defined as having indeterminate cytology with the risk of malignancy ranging from 10%∼75%. There are three categories of classification for these malignancies; these include the Bethesda category III (atypia of undetermined significance or follicular lesion of undetermined significance, AUS/FLUS), Bethesda category IV (follicular neoplasm/ suspicious for a follicular neoplasia, SFN/FN), and Bethesda category V (suspicious for malignancy, SFM) (Baloch et al., 2018). Pathologists tend toward the conservative management of this disease, lending itself to conflicts in the physician-patient relationship and moderate reproducibility of cytologically indeterminate samples (Sauter et al., 2019). Furthermore, the performance of interventional radiologists in providing ultrasound-guided thyroid fine needle aspirations (FNAs) and the experience of endocrinologists and cytopathologists in interpreting aspirates varies among institutions or hospitals (Sauter et al., 2019; Bose, Sacks & Walts, 2019). These may result in repeat FNA, diagnostic lobectomy, and even unnecessary total thyroidectomy procedures. Thus, molecular pathogenesis has become a promising approach to supplement cytologic examination for preoperative diagnosis and treatment options (Ren et al., 2022; Lee et al., 2022).

Diverse molecular testing panels have been developed based on next-generation sequencing (NGS) techniques. For instance, ThyroSeq, ThyGenX and ThyraMIR tests have both high PPV and NPV and can be used to determine malignancy, while Afirma GSC tests show high NPV and relatively low PPV and might be useful to rule-out the presence of malignancy for indeterminate thyroid lesions (Bose, Sacks & Walts, 2019; Lee et al., 2022; Rossi et al., 2022). There is limited research applying these tests in the Chinese population and few verified self-designed multigene panels are available on the market (Ren et al., 2022; Song et al., 2020). Here, we designed a multigene NGS panel to detect targeted DNA alterations and RNA fusions. We retrospectively evaluated its performance in FNA specimens of cytologically indeterminate thyroid nodules and compared these results to those of a clinical diagnosis. Our study proposed an auxiliary diagnosis to reduce cytologically diagnostic uncertainty and to explore the possible application of molecular risk stratification based on our NGS panel. Furthermore, this study isolated nodules with concurrent molecular alterations from single aberrations and analyzed the two sets of data separately.

Materials & Methods

Study cohorts and sample collection

Patients with at least one thyroid nodule that was clinically diagnosed and confirmed by ultrasound were retrospectively enrolled and screened in our study (3,175 cases). Patients under 18 years old, who did not have a primary diagnosis, or who were missing the required information (2,343 cases) were excluded, as detailed in Fig. 1. Fine-needle aspiration (FNA) samples were collected under ultrasound guidance by radiologists using a 22-gauge needle. The aspirate was smeared on microscope slides and stained after being fixed with 95% ethanol for cytological examination. Categorization was conducted according to The Bethesda System for Reporting Thyroid Cytopathology (TBSRTC). A total of 529 FNA samples from cytologically indeterminate thyroid nodules (i.e., diagnosed by clinical pathologists as Bethesda III, IV or V) were collected to analyze the DNA mutations and RNA fusions via NGS. The clinical diagnosis of the nodules and their subsequent follow-up reports following definitive surgery, if performed (Table 1) were analyzed retrospectively for further investigation. This retrospective study was approved by the Ethics Committee of Jiangsu Integrated Traditional Chinese and Western Medicine Hospital (No. 2022-LWKY-042) and the Ethics Committee of Guangzhou KingMed Medical Laboratory Center (No. 2023005). Patient consent was waived due to the observational nature of the retrospective study and patient identities were kept anonymous (No. 2023005).

Figure 1 Recruitment and exclusion of patients and samples in the study.

Table 1 Baseline characteristics of thyroid nodules with molecular pathology results (n, column %).

Characteristics	Molecular pathology	χ 2	P -value	
	Benign nodules (n = 34)	Suspicious nodules (n = 495)			
Age at diagnosis					
Under 55 years	21 (61.76)	403 (81.41)	7.721	0.005	
55 years and older	13 (38.24)	92 (18.59)			
Gender					
Female	28 (82.35)	391 (78.99)	0.218	0.64	
Male	6 (17.65)	104 (21.01)			
Cytologic classification					
Bethesda III nodules	6 (17.65)	71 (14.34)	38.829	<0.001	
Bethesda IV nodules	12 (35.29)	30 (6.06)			
Bethesda V nodules	16 (47.06)	394 (79.60)			
Thyroid surgery					
Surgery	30 (88.24)	488 (98.59)	45.762	<0.001	
No surgery/Unknown	4 (11.76)	7 (1.41)			

Molecular analysis

Sequencing libraries were generated using one-step multiplex PCR targeted amplicons. Briefly, the genomic DNA were isolated from the FNA samples of thyroid nodules using the QIAamp DNA Mini kit (Qiagen, Hilden, Germany) in accordance with the manufacturer’s specifications. The concentrations were determined by Qubit 3.0 Fluorometer (Thermo Fisher Scientific, Waltham, MA, USA). Sample DNA were mixed with KAPA2G Fast PCR (Roche, La Posay, France) and were amplified to detect driver gene variants, utilizing our multigene testing panel (Table S1) for the detection of 14 thyroid cancer-related genes (AKT1, BRAF, CTNNB1, EIF1AX, HRAS, KRAS, NRAS, PAX8, PIK3CA, PTEN, RET, TERT, THADA, and TP53) and 21 types of gene rearrangements occurring in thyroid cancer (ACBD5, AFAP1L2, ALK, ATG10, BRAF, CALM2, CCDC6, ERC1, ETV6, FLNC, FMNL2, KIAA1217, KIAA1594, KIF20B, NCOA4, NTRK3, PAK1, PAX8, PIBF1, PPAR γ, PXK, RALGAPA2, RET, SND1, and STRN). Superfluous primers were purified using Agencourt AMPure XP 60 mL kit (Beckman, Brea, CA, USA). The concentrations of barcoded PCR produced library were then measured and diluted to 100 pM. A total of 20 µL pooled amplicons were sequenced at the Ion Proton system (Thermo Fisher Scientific, Waltham, MA, USA). Local alignments of reads to the hg19 genome were performed via Bowtie2 (version 2.2.4) in the paired-end mode. SAM alignment files were converted to BAM files, sorted, and indexed using Samtools (version 0.1.19), following by a procession with Bam-read count and a customized written Perl script.

Test performance evaluation and false-positive/-negative (FP/FN) results analysis

The actual status of each diagnosis was determined: (1) by surgical pathology or (2) a benign molecular test without surgical pathology. The results of the benign molecular test are only considered truly benign when they are based on ultrasound and/or clinical characteristics. Test performance characteristics including sensitivity, specificity, positive predictive value (PPV), negative positive predictive value (NPV), and Kappa-value for consistency check were calculated at 95% confidence interval (CI) for the overall cytologically indeterminate specimens, as well as separately for nodules in Bethesda III, IV and V following the established method (Altman & Bland, 1994). Based on observed sensitivity and specificity, hypothetical PPV and NPV curves were modeled over the entire range of possible disease prevalence (0–100%), allowing the observed and the anticipated PPV and NPV to be compared. In order to preclude misdiagnosis due to false-negative test results, histologic slides of nodules that had negative molecular results on FNA cytology but had been diagnosed as malignant on resection, were blindly reviewed by another pathologist.

Molecular risk stratification

Thyroid nodules with molecular aberrations were categorized into three molecular risk groups (MRGs): a high-risk group (TERT or TP53 alterations), a BRAF-like group (mainly BRAFV 600E) and a RAS-like group (HRAS, KRAS, NRAS, and others). The classification of all variants into this three-category system is shown in Table S2. A sample would be considered positive for molecular pathology if some genetic alteration was detected; the sample would be negative if no variants were detected in our panel. The MRG results would be assessed for the relevance with clinicopathologic diagnosis and further analyzed for the aggressive features of tumor-node-metastasis (TNM) staging. Nodules with concurrent molecular alterations were isolated and their distribution in MRGs and their TNM classification were assessed individually. Slides of surgical histopathology for special cases (nodules suspected as false-negative or false-positive, presenting concurrent molecular alterations, and other infrequent genetic alterations) were retrospectively analyzed by another pathologist.

Statistical analyses

Descriptive summaries of the histopathology and molecular pathology reports were shown with counts and percentages. Pearson chi-square tests were used to compare the categorical variables of age, gender, Bethesda category, and thyroid surgery between benign/suspicious molecular result groups. Statistical analysis was conducted utilizing SPSS software (version 26.0) with P-value < 0.05 being considered as statistically significant.

Results

Study cohort and demographics

A total of 529 cases with cytologically indeterminate thyroid FNA results were identified between February 2021 to December 2021 at Jiangsu Province Hospital on Integration of Chinese and Western medicine in China. These cases were analyzed for molecular testing via NGS. All data in the cohort are accessible in Table S2. The patient characteristics and thyroid nodules are shown in Table 1. A total of 391 (79.88%) patients with suspicious nodules as determined by molecular tests were female and 403 (81.41%) of the cases for the cohort were below 55 years old at time of diagnosis. The majority of thyroid nodules (410 cases, 77.50%) were interpreted as Bethesda V, and 14.56% (77 cases), and 7.94% (42 cases) were Bethesda III and IV, respectively. Overall, the results of our thyroid NGS panel determined benign nodules (34 cases, 6.43%) and suspicious nodules (495 cases, 93.57%). The accuracy of the test was checked to evaluate the performance of the multigene panel. Furthermore, 574 molecular aberrations with 89 samples containing coexisting nodules were further investigated for their associated histopathologic cancer types and TNM staging based on molecular risk stratification (Fig. 1).

Test performance

The test performance for sensitivity, specificity, NPV, PPV, and consistency of the cytologic groups of thyroid nodules is presented in Table 2. Since sensitivity and specificity are intrinsic characteristics for each test, the NPV and PPV depended on the prevalence of the disease in the screened population (Smith, 2012). Predicated PPV and NPV with 95% CI were calculated based on the observed sensitivity and specificity; hypothetical PPV and NPV curves were modeled over the entire range of possible disease prevalence (0–100%), as depicted in Fig. 2.

Table 2 Performance of the molecular test in cytologically indeterminate thyroid nodules.

Performance in Bethesda III nodules (n = 77; disease prevalence 90.91%)	
Molecular test	Histopathologic diagnosis	Test performance, % (95% CI)	Consistency check	
Positive (n = 70)	Negative (n = 7)			
Positive	69	2	PPV, 97.18 (91.45–99.11)	Sensitivity, 98.57 (92.30–99.96)	Kappa = 0.748	
Negative	1	5	NPV, 83.33 (40.32–97.37)	Specificity, 71.43 (29.04–96.33)	Substantial agreement	
Performance in Bethesda IV nodules (n = 42; disease prevalence 66.67%)	
Molecular test	Histopathologic diagnosis	Test performance, % (95% CI)	Consistency check	
Positive (n = 28)	Negative (n = 14)			
Positive	27	3	PPV, 90.00 (76.70–96.09)	Sensitivity, 96.43 (81.65–99.91)	Kappa = 0.778	
Negative	1	11	NPV, 91.67 (61.16–98.72)	Specificity, 78.57 (49.20–95.34)	Substantial agreement	
Performance in Bethesda V nodules (n = 410; disease prevalence 98.29%)	
Molecular test	Histopathologic diagnosis	Test performance, % (95% CI)	Consistency check	
Positive (n = 403)	Negative (n = 7)			
Positive	394	0	PPV, 100.00 (–)	Sensitivity, 97.77 (95.80–98.97)	Kappa = 0.599	
Negative	9	7	NPV, 43.75 (28.96–59.74)	Specificity, 100.00 (59.04–100.00)	Moderate agreement	
Performance in Bethesda III, IV and V nodules (n = 529; disease prevalence 94.71%)	
Molecular test	Histopathologic diagnosis	Test performance, % (95% CI)	Consistency check	
Positive (n = 501)	Negative (n = 28)			
Positive	490	5	PPV, 98.99 (97.79–99.54)	Sensitivity, 97.80 (96.11–98.90)	Kappa = 0.726	
Negative	11	23	NPV, 67.65 (53.20–79.36)	Specificity, 82.14 (63.11–93.94)	Substantial agreement	

Figure 2 Predicated performance of molecular test in populations with different disease prevalence.

Predicated PPV (solid orange lines) and NPV (solid blue lines) with 95% CI (dotted lines) based on sensitivity and specificity for: (A) the overall cytologically indeterminate specimens (Bethesda III, IV and V); (B–D) Bethesda III, IV and IV cytology thyroid nodules separately. NPV and PPV in the expected range of cancer/NIFTP prevalence (green rectangle) based on the malignant risk of different categories by Bethesda system were shown. PPV, positive predictive value; NPV, negative predictive value; CI, confidence interval; NIFTP, noninvasive follicular thyroid neoplasm with papillary-like nuclear features.

Among the positive test samples, 490 cases (98.99%) were malignant and five cases (1.01%) were benign on surgery resection, while 23 cases (67.65%) were benign and 11 cases (32.35%) were cancerous within the negative test samples. Overall, in the cohort of cytologically indeterminate nodules had a cancer prevalence of 94.71%; multigene testing presented a sensitivity of 97.80% (95% CI [96.11–98.90]%), a specificity of 82.14% (95% CI [63.11–93.94]%), a PPV of 98.99% (95% CI [97.79–99.54]%) and an NPV of 67.65% (95% CI [53.20–79.36]%) (Table 2). The tumor prevalence in our cohort was obviously higher than clinical actuality due to the difficulty of identifying “true” benign nodules without surgery, risks of malignancy (with NIFTP) published in the TBSRTC diagnostic categories were referenced as the expected prevalence range (marked with green rectangles in Fig. 2). Moreover, the consistency of the multigene testing performed better in Bethesda III and IV (Kappa = 0.748 and 0.778, respectively, ranked as substantial agreement) than V (Kappa = 0.599, ranked as moderate agreement) specimens. When three categories of cytologically intermediate nodules were pooled, the consistency check for molecular test with histopathologic diagnosis manifested substantial agreement (Kappa = 0.726) (Table 2).

Summary of molecular alterations

All the somatic single nucleotide variants (SSNVs), insertions, deletions, duplications, and fusions detected in cytological indeterminate thyroid nodules were summarized in Table 3. The most observed gene aberration in this cohort was BRAFV 600E (446 cases, 82.59%), followed by RET aberrations (20 fusions and two mutants, 4.07%), NRAS mutants (22 cases, 4.07%), KRAS mutants (17 cases, 3.15%), and other alterations. Notably, CCDC6-RET fusion was detected in 17 cases, which may be related to 16 patients with papillary thyroid carcinoma or microcarcinoma (PTC/PTMC) and one benign case with concurrent molecular alteration. Moreover, some molecular aberrations may co-exist with others rather than appear alone; these include BRAF rearrangements and deletions, PTEN deletions, TERTQ302R, and mutants of AKT1, EIF1AX, TP53, and PIK3CA. There are some molecular alterations that had never been reported in solid tumors before, including BRAF-PXK fusion, FLNC-BRAF fusion, BRAFN486_T491delinsS, PAX8C147R, PTEND252Rfs∗42, and TERTQ302R, as well as the EIF1AX c.338-2A>T splice site mutation and the KIAA1217-RET fusion in thyroid cancers.

Table 3 Summary of molecular alterations in cytologically indeterminate thyroid nodules.

Gene	Variant a	Alteration (Count)	Count	% of Total	Referene	
AKT1	SSNV	exon4 NM_001014432.1: c.155T>G (p.L52R)	2	0.37%		
	SSNV	exon4 NM_001014432.1: c.49G>A (p.E17K)	1	0.19%		
ALK	fusion	STRN-ALK	1	0.19%	Chu et al. (2020)	
BRAF	SSNV	exon15 NM_004333.4: c.1799T>A (p.V600E)	446	82.59%		
SSNV	exon15 NM_004333.4: c.1801A>G (p.K601E)	1	0.19%		
indel	exon12 NM_004333.4: c.1456_1471delinsT (p.N486_T491delinsS)	1	0.19%		
focal deletion	exon12 NM_004333.4: c.1457_1471del (p.N486_P490del)	1	0.19%		
fusion	BRAF-PXK	1	0.19%		
fusion	FLNC-BRAF	1	0.19%		
fusion	SND1-BRAF	1	0.19%	Chu et al. (2020)	
EIF1AX	SSNV	exon6 NM_001412.3: c.338-2A>T (p.?)	1	0.19%	Castagna et al. (2020)	
HRAS	SSNV	exon2 NM_005343.2: c.34G>A (p.G12S)	1	0.19%		
SSNV	exon3 NM_005343.2: c.181C>A (p.Q61K)	1	0.19%		
SSNV	exon3 NM_005343.2: c.182A>G (p.Q61R)	1	0.19%		
SSNV	exon3 NM_005343.2: c.182A>T (p.Q61L)	1	0.19%		
KRAS	SSNV	exon3 NM_004985.3: c.181C>A (p.Q61K)	4	3.15%		
SSNV	exon2 NM_004985.3: c.35G>A (p.G12D)	3	0.56%		
SSNV	exon2 NM_004985.3: c.35G>T (p.G12V)	3	0.56%		
SSNV	exon2 NM_004985.3: c.38G>A (p.G13D)	2	0.37%		
SSNV	exon3 NM_004985.3: c.182A>G (p.Q61R)	1	0.19%		
SSNV	exon4 NM_004985.3: c.436G>A (p.A146T)	1	0.19%		
SSNV	exon4 NM_004985.3: c.437C>T (p.A146V)	1	0.19%		
SSNV	exon2 NM_004985.3: c.34G>A (p.G12S)	1	0.19%		
SSNV	exon2 NM_004985.3: c.37G>C (p.G13R)	1	0.19%		
NRAS	SSNV	exon3 NM_002524.4: c.182A>G (p.Q61R)	17	3.15%		
SSNV	exon2 NM_002524.4: c.35G>A (p.G12D)	2	0.37%		
SSNV	exon3 NM_002524.4: c.181C>A (p.Q61K)	2	0.37%		
SSNV	exon2 NM_002524.4: c.34G>T (p.G12C)	1	0.19%		
PAX8	SSNV	exon5 NM_003466.3: c.439T>C (p.C147R)	1	0.19%		
SSNV	exon7 NM_003466.3: c.659G>A (p.R220Q)	1	0.19%		
PIK3CA	SSNV	exon10 NM_006218.2: c.1624G>A (p.E542K)	1	0.19%		
SSNV	exon10 NM_006218.2: c.1633G>A (p.E545K)	1	0.19%		
SSNV	exon12 NM_006218.2: c.1850G>A (p.R617Q)	1	0.19%		
SSNV	exon21 NM_006218.2: c.3140A>G (p.H1047R)	1	0.19%		
SSNV	exon21 NM_006218.2: c.3140A>T (p.H1047L)	1	0.19%		
PTEN	SSNV	exon5 NM_000314.4: c.323T>C (p.L108P)	1	0.19%		
focal deletion	exon5 NM_000314.4: c.437del (p.L146*)	1	0.19%		
focal deletion	exon7 NM_000314.4: c.754_764del (p.D252Rfs*42)	1	0.19%		
RET	fusion	CCDC6-RET	17	3.15%	Chu et al. (2020) Lee et al. (2016)	
fusion	NCOA4-RET	2	0.37%	
fusion	KIAA1217-RET	1	0.19%		
SSNV	exon16 NM_020975.4: c.2753T>C (p.M918T)	1	0.19%	Subbiah et al. (2021); Ghazani et al. (2020); Alzahrani et al. (2022)	
SSNV	exon14 NM_020975.4: c.2410G>A (p.V804M)	1	0.19%	
TERT	SSNV	exon2 NM_198253.2: c.905A>G (p.Q302R)	1	0.19%		
TP53	SSNV	exon10 NM_000546.5: c.1073A>T (p.E358V)	1	0.19%		
SSNV	exon7 NM_000546.5: c.731G>A (p.G244D)	1	0.19%		
SSNV	exon7 NM_000546.5: c.734G>T (p.G245V)	1	0.19%		
SSNV	exon7 NM_000546.5: c.742C>T (p.R248W)	1	0.19%		
focal deletion	exon4 NM_000546.5: c.245_246del (p.P82Rfs*66)	1	0.19%		
focal deletion	exon8 NM_000546.5: c.783-806del (p.G262-S269del)	1	0.19%		
duplication	exon4 NM_000546.5: c.273dup (p.P92Afs*57)	1	0.19%		
Total			540	100%		

False-negative and false-positive molecular test results with histopathologic diagnosis

The clinicopathologic characteristics of 11 false-negative (Case 1 to Case 11) and five false-positive (Case 12 to Case 16) cases are listed in Table 4. In total, 11 malignant nodules tested negative in our cohort, including one in Bethesda III, one in Bethesda IV, and nine in Bethesda V. These patients underwent surgery due to other suspicious features, subject to their preference. All of the thyroid nodules proved to be PTC/PTMC after diagnostic surgery, in which the presence of extrathyroidal invasion (T4a_staging) was observed in three cases. Four cases revealed locoregional nodal metastases (N1a/b_staging). Two nodules that tested negative before surgery (Case 1 and Case 4) were verified positive with the BRAFV 600E mutant on resection, indicating that molecular misdiagnosis may be the result of a limited tumor volume, sampling technique, or preservation method.

Table 4 Clinicopathologic characteristics of cases with false-positive (FP) or false-negative (FN) test results.

Case No.	Age (year)	Gender (M/F)	Nodule size (cm) a	Cytology diagnosis	Histological diagnosis	T_stage	N_stage	Molecular aberrations	Other indicators	Disagreement	
1	63	F	1.3	Bethesda V	PTC	T1b	N0	Undetected	ARMS-PCR: BRAFV 600E (+)	FN	
2	50	F	1.1	Bethesda V	PTC	T1b	N1a	Undetected	–	FN	
3	55	F	1.0	Bethesda V	PTC	T1a	N1a	Undetected	–	FN	
4	46	F	0.7	Bethesda IV	PTMC	T1a	N0	Undetected	ARMS-PCR: BRAFV 600E (+)	FN	
5	66	F	0.7	Bethesda V	PTMC; Lymphocytic thyroiditis	T4a	N1b	Undetected	LN-FNA-T g > 500.000 ng/mL	FN	
6	22	F	0.7	Bethesda V	PTMC	T4a	N0	Undetected	–	FN	
7	50	F	0.6	Bethesda III	PTMC	T1a	N1b	Undetected	–	FN	
8	48	M	0.5	Bethesda V	PTMC	T1a	N0	Undetected	–	FN	
9	45	F	0.5	Bethesda V	PTMC	T4a	N0	Undetected	–	FN	
10	40	F	0.5	Bethesda V	PTMC	T1a	N0	Undetected	–	FN	
11	53	F	0.3	Bethesda V	PTMC	T1a	Nx	Undetected	–	FN	
12	37	F	3.0	Bethesda IV	Follicular adenoma (FA)	NA	NA	BRAF V 600E	NA	FP	
13	62	M	NA	Bethesda III	Lymphocytic thyroiditis	NA	NA	BRAF V 600E	NA	FP	
14	49	F	NA	Bethesda III	Nodular goiter with adenomatoid nodule	NA	NA	BRAF V 600E	NA	FP	
15	55	F	NA	Bethesda IV	Nodular goiter with adenomatoid nodule; Lymphocytic thyroiditis	NA	NA	NRASQ61R; KRASA146V	NA	FP	
16	53	F	NA	Bethesda IV	Hashimoto’s thyroiditis (HT) with adenomatoid nodule	NA	NA	BRAFV 600E; CCDC6-RET	NA	FP	
Notes.

a Measured in pathological specimen.

The false-negative result of Case 5 may be specious, since the value of FNA-Tg was significantly high (above 500.000 ng/mL) in the puncture fluid of the left lymph nodule and FNAC manifested chronic lymphocytic thyroiditis. However, none of the molecular aberrations were detected in either the thyroid and lymph nodules at diagnosis. Similarly, the false-positive result of Case 12 may require further study for a potentially malignant transformation, given that the final histological diagnosis was benign follicular adenoma (FA), three cm in diameter. This nodule was suspected as being follicular neoplasia by FNA cytology and the BRAFV 600E mutant was detectable before surgery.

Molecular risk stratification (concurrent molecular alterations excluded)

The cancer subtypes in specific molecular alterations and MRGs are summarized in Table 5. After the exclusion of concurrent molecular alterations, only two nodules with TP53 alterations were divided into the high-risk group, one (TP53P82Rfs∗66) of which was follicular thyroid carcinoma (FTC) with partial differentiation. The other (TP53G262−S269del) was determined to be PTMC on resection, however it was initially cytologically categorized as Bethesda IV (suspicious for a follicular neoplasia). The prevalence of PTC/PTMC in the BRAF-like group reached 100% in 426 tumors, while 22 malignant nodules in the RAS-like group were composed of 14 PTC/PTMC (63.64%) and eight FTC or follicular patterned tumors (follicular variant PTC/PTMC, FV-PTC/FV-PTMC) (36.36%).

Table 5 Probability of cancer in specific molecular alterations and classifications (concurrent molecular alterations excluded).

Molecular classification	Prevalence in test-positive samples, No. (%)	Molecular alterations, No.	Histopathologic diagnosis, No.	Cancer type	Cancer type, No. (%)	
				Benign	FTC	FV-PTMC	FV-PTC	PTMC	PTC			
High-risk group	2 (0.44)									PTC/PTMC	1 (50)	
	TP53	2	0	1	0	0	1	0	FV-PTC/FV-PTMC/FTC	1 (50)	
BRAF-like group	426 (94.67)											
	V600E-BRAF	407	3	0	0	0	135	269	PTC/PTMC	423(100)	
	CCDC6-RET	15	0	0	0	0	4	11	FV-PTC/FV-PTMC/FTC	0 (0)	
	KIAA1217-RET	1	0	0	0	0	1	0			
	NCOA4-RET	2	0	0	0	0	0	2			
	RET	1	0	0	0	0	1	0			
RAS-like group	22 (4.89)											
	non-V600E-BRAF	2	0	0	0	0	0	2	PTC/PTMC	14 (63.64)	
	HRAS	1	0	0	0	0	0	1	FV-PTC/FV-PTMC/FTC	8 (36.36)	
	KRAS	6	0	0	0	0	0	6			
	NRAS	9	0	3	2	1	0	3			
	PAX8	2	0	0	1	0	1	0			
	PTEN	1	0	1	0	0	0	0			
	STRN-ALK	1	0	0	0	0	1	0			
Undetected	NA									PTC/PTMC	11 (100)	
		—	34	23	0	0	0	9	2	FV-PTC/FV-PTMC/FTC	0 (0)	
Total	450 (100)		484	26	5	3	1	153	296			

TNM staging and aggressive features (concurrent molecular alterations excluded)

The extrathyroidal extension and TNM stage of thyroid carcinomas were proposed to be associated with molecular risk subtyping (Hong et al., 2022). Thus, we investigated the aggressive features of the molecular alterations via the categorization of MRGs in 485 nodules with known TNM staging information (data displayed in Table S2). Among 174 BRAF-like tumors with confirmed diagnosis (PTC or PTMC) and Tx staging excluded, 57 cases (32.76%) were observed infiltration of thyroid capsule (T4a or T4b staging) and 115 cases (46.56%) of 247 assessable data (diagnosis of PTC or PTMC and Nx staging excluded) happened lymph node metastases (N1a or N1b staging). In RAS-like groups, however, two (22.22%) of the nine malignant nodules and four (40.00%) of the 10 cases were observed to have infiltration of the thyroid capsule and lymph node metastases, respectively. The dominating aberrations may be the BRAFV 600E mutant and deletion-insertions, along with RET mutants and fusions.

Concurrent molecular alterations and histopathological characteristics

A total of 44 nodules with concurrent molecular alterations were isolated and separately analyzed to determine whether some aberrations tended to co-occur with others and present different histopathological characteristics. Figure 3 shows that all cases were classified as PTC or PTMC, with the exception of two benign nodules and one FTC. The most common BRAFV 600E could be detected in 39 nodules (88.64%) concurrently with other 25 mutants from eight genes, including TERT, TP53, PIK3CA, AKT1, BRAFK601E, HRAS, KRAS, and NRAS, followed by the NRASQ61R mutant, which occurred in nine nodules (20.45%). Among the remaining five cases without the BRAFV 600E mutant, two PTC nodules were related to the BRAF rearrangements with the PXK, FLNC, or SND1 gene, while one PTC nodule underwent double deletions in the PTEN gene and one benign nodule revealed concurrent mutants in KRASA146V and NRASQ61R. An unknown mutant in the protein detected in EIF1AXc.338−2A>T was concurrent with HRASQ61R in a FTC nodule. These findings agreed with previous reports (Fagin & Wells Jr, 2016).

Figure 3 Histopathologic diagnosis and TNM staging in thyroid nodules with concurrent molecular alterations.

Previous studies have focused on the relationship of molecular markers with distant metastases, such as the BRAFV 600E mutation and TERTp mutations. These were frequently found to co-exist and their presence was considered to be valuable for PTC relapse risk assessment (Ulisse et al., 2021; Soares et al., 2021). Likewise, in our cohort, potentially aggressive nodules may also be related to concurrent molecular alterations involving BRAFV 600Ebeing accompanied by TERTQ302R, AKT1L52R, NRASG12C, NRASQ61R and CCDC6-RET fusion. Five cases with lymph node metastases (N1 staging, marked with triangle symbols in Fig. 3) presented with infiltration of the thyroid capsule (T4 staging, marked with pentagram symbols in Fig. 3) after surgery resection, although one nodule with concurrent mutations of BRAFV 600E and NRASG12C was an exception. Histopathologic slides of a resection in two cases of FV-PT(M)C were reviewed and the photomicrographs are presented in Fig. 4. The nodule with the combination mutations of BRAFV 600E, TP53G244D and KRASQ61R were poorly differentiated, while the lesion with concurrent alterations of BRAFV 600E and TERTQ302R displayed an extrathyroidal extension with regional nodal involvement and local invasion of the skeletal muscle (data not shown) and adipose tissue (Fig. 4). These observations indicated that thyroid nodules with concurrent molecular alterations may be related with poor differentiation, aggressiveness, and a follicular variant of the PTC subtype.

Figure 4 Photomicrographs from 2 cases of FV-PT(M)C with concurrent molecular alterations.

In the first case (A–C), molecular testing revealed BRAFV600E co-existence with TP53 G244D and KRASQ61R. Tumor cells arranged in solid nests and showed classical nuclear features of PTC including several intranuclear pseudoinclusions (arrows in C) though being diagnosed as FV-PTMC with poorly differentiated appearance. In the second case (D–F), molecular testing revealed BRAFV600E co-existence with TERTQ302R. The nodule was classified as FV-PTC with calcification, and extrathyroidal extension was present (E) with locally invading adipose tissue. H&E staining. (A and D) 100 × magnification; (B and E) 200 × magnification; (C and F) 400 × magnification. FV-PT(M)C, follicular variant subtype of papillary thyroid carcinoma or microcarcinoma.

Discussion

A multigene panel for cytologically indeterminate thyroid nodules was designed, and molecular results were retrospectively assessed for their performance when compared to a clinical diagnosis. The overall consistency was acceptable with high NPV which made the panel a promising tool to rule out the malignant Bethesda category III and IV nodules. Positive tests in the Bethesda V nodules suggested treatment with a total thyroidectomy instead of a lobectomy. Eleven false-negative and five false-positive cases were found in the cohort. During the analysis of the false negatives, the presence of low-frequency mutations that were not highlighted by the automatic analysis software IGV were checked. However, false negatives may have occurred due to: (1) a molecular misdiagnosis caused by limited tumor volume or an improvable sampling technique and preservation method; (2) molecular alterations in genes that occurred beyond the scope of our panel (namely, the detection of 14 thyroid cancer-related genes and 21 types of gene rearrangements). The false positives may have occurred for a number of reasons, including limited clinical follow-up. Additionally, the extremely high prevalence of cancer in the cohort should be noted for clinical reference of multigene testing in presurgical diagnosis of thyroid nodules.

The most observed gene aberration in our cohort referred to driving somatic genetic alterations in the MAPK pathway, including the BRAFV 600E mutant, RET aberrations, and RAS mutations. Furthermore, several molecular aberrations were observed in thyroid cancers or even solid tumors for the first time. There were some infrequently reported alterations, including EIF1AX c.338-2A>T splice site KIAA1217-RET fusion, that were considered to be pathogenetic mutations or oncogenic driver genes for malignant tumors, as revealed in previous research (Castagna et al., 2020; Elsherbini et al., 2022; Lee et al., 2016; Song et al., 2022; Davis et al., 2020; Lee et al., 2016). These new findings may support genetic profiling and could be potential therapeutic targets for thyroid cancers.

For the purpose of ancillary diagnosis and outcome prediction, the malignancy risk of thyroid nodules could be stratified based on different molecular aberrations (Soares et al., 2021; Yip et al., 2021; Skaugen et al., 2022). In our study, detected molecular aberrations were categorized into three MRGs: the BRAF-like group (presenting low differentiation and predominantly relevant to classical PTC with papillary architecture), the RAS-like group (displaying high differentiation, less recurrence, and enriched in thyroid carcinomas with follicular-pattern), and the high-risk group (relevant to poorly clinical outcomes and typically coexisting with other alterations) (Cancer Genome Atlas Research, 2014).

The BRAFV 600E and mutated RAS were considered to be two mutually exclusive drivers of PTC, possibly suggesting similar or redundant downstream effects, and causing different signaling effects followed by profound phenotypic differences (Cancer Genome Atlas Research, 2014; Ren et al., 2022) However, less attention had been paid to concurrent molecular alterations, conveying a significant adverse prognosis and suggesting treatment via a total thyroidectomy of the thyroid nodules (Bose, Sacks & Walts, 2019; Ren et al., 2022; Poller & Glaysher, 2017). In our study, some drivers in the PI3K pathway and tumor suppressor genes manifested a tendency to be concurrent with others, such as BRAF rearrangements and deletions, PTEN deletions, TERTQ302R, and mutations of AKT1, EIF1AX, TP53, and PIK3CA. As shown in Fig. 3, 26 cases (59.09%) in all 44 nodules with concurrent molecular alterations were detected concomitant BRAF/RAS mutation. Besides, the nodules that indicated the presence of TERT or TP53 mutants concurrent with BRAF-like mutations presented the follicular variant of the PTC subtype and were indicative of poor differentiation or aggressive features. The more aggressive features and poorer progression of the BRAF-like group versus the RAS-like group were consistent with previous reports (Rossi et al., 2022; Krasner et al., 2019).

Bioinformational tools have found that BRAF mutants or fusions were strongly separated from the H/N/KRAS mutants and partly overlapped the RET fusions in clusters (Cancer Genome Atlas Research, 2014). However, in our cohort, BRAFV 600E was found to be a co-mutant with RAS and PIK3CA mutations, while the RET fusions were exclusive with the BRAF aberrations. These ambiguous discrepancies may have resulted from an unverified computerized algorithm, racial differences, or insufficient sample sizes, etc. The types of mutations in the oncogenes or tumor suppressor genes could also determine the biological behavior of malignant tumors (Kelil et al., 2016). In addition to concurrent alterations and mutation types, diverse mutations in the same genes resulting in variants of identical or adjacent codons might be related to different biologic behaviors. For example, HRASQ61K was detected in one PTC, HRASQ61Lco-mutant with BRAFV 600E in one PTC, while HRASQ61R was detected as a co-mutant with the EIF1AX mutant in one FTC; KRASA146T or NRASQ61Kwas a detectable co-mutant with BRAFV 600Ein PTCs, though KRASA146V and NRASQ61R were found to be concurrent in a benign nodule. These assumptions appealed to the further classification of molecular alterations and conjoint analysis with pathological results for clinical reference.

Some limits were present in the study: (1) the limitation presented by the extremely high prevalence of cancer in the cohort should be noted, as well as the low sample size of Bethesda III and IV groups; (2) a well-designed prospective study with informed consent of the patients will be essential for more informative results in the future; (3) the long-term follow-up above five years may be beneficial to further summarize the prediction values for remote metastasis and disease outcomes and to elucidate the outcomes of unresected indeterminate nodules that prospectively underwent analysis of the panel in this study.

Conclusions

The successful presurgical diagnosis of thyroid nodules is debatable. Although molecular tests such as Thyroseq or Afirma have been demonstrated to have a very good performance, they are not readily available. Treatment facilities are often left to find their own resources to identify those indeterminate nodules that need to be surgically treated. Our study reported the results of a multi-gene panel analysis on cytological samples from indeterminate thyroid nodules and proposed a model for risk stratification that was able to predict cases that should then be referred to surgery. However, the extremely high prevalence of cancer in the studied cohort was a limitation of our study.

Supplemental Information

Supplemental Information 1 List of multigene testing panel

Click here for additional data file.

Supplemental Information 2 All diagnosis data in the cohort

Click here for additional data file.

We would like to thank the pathologists and surgeons of the Affiliated Hospital of Integrated Traditional Chinese and Western Medicine (Nanjing, P.R. China) for their assistance with the pathological diagnoses of the surgical resections and FNA samples.

Additional Information and Declarations

Competing Interests

Author Contributions

Ethics

Data Availability

Yuanyuan Zhou, Zhiqiang Li, Xia Ge, Hao Chen and Yuan Mao are the employees of KingMed Center for Clinical Laboratory Co., Ltd, Hefei, Anhui Province, P.R. China. The remaining authors declare that that they have no competing interests.

Yuanyuan Zhou conceived and designed the experiments, performed the experiments, analyzed the data, prepared figures and/or tables, authored or reviewed drafts of the article, and approved the final draft.

Xinping Wu conceived and designed the experiments, analyzed the data, authored or reviewed drafts of the article, and approved the final draft.

Yuzhi Zhang performed the experiments, analyzed the data, prepared figures and/or tables, and approved the final draft.

Zhiqiang Li conceived and designed the experiments, prepared figures and/or tables, and approved the final draft.

Xia Ge performed the experiments, prepared figures and/or tables, and approved the final draft.

Hao Chen performed the experiments, prepared figures and/or tables, and approved the final draft.

Yuan Mao conceived and designed the experiments, authored or reviewed drafts of the article, and approved the final draft.

Wenbo Ding conceived and designed the experiments, analyzed the data, authored or reviewed drafts of the article, and approved the final draft.

The following information was supplied relating to ethical approvals (i.e., approving body and any reference numbers):

the Ethics Committee of Jiangsu Integrated Traditional Chinese and Western Medicine Hospital

The following information was supplied regarding data availability:

The raw measurements are available in the Supplementary Files.

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
