# Peer review of "Performance of multigene testing in cytologically indeterminate thyroid nodules and molecular risk stratification"

_PeerJ, doi:10.7717/peerj.16054_

## Round 0.1 · original submission · Major Revisions

Dear Dr. Ding,

Your manuscript entitled “Performance of a multigene testing in cytologically indeterminate thyroid nodules and molecular risk stratification" which you submitted to PeerJ, has been reviewed by the editor and 3 external reviewers.

I regret to inform you that the reviewers have raised some significant concerns. In light of these concerns, further editorial consideration would not be possible without extensive and substantive revisions. In particular, the issue regarding the validity of the findings must be adequately addressed. The reviewers’ comments are reported below.

If you decide to resubmit the revised version, please summarize all the improvements made in the new version and give answers to all critical points raised in the reviewers’ report in an accompanying letter. Copy and paste each and every reviewer's comment above your response.

Please note that resubmitting your manuscript does not guarantee eventual acceptance. Since the requested changes are major, the revised manuscript will undergo a second round of review by the same reviewers. I must emphasize that the acceptability of the revision will depend upon the resolution of the points raised by the reviewers.

Sincerely yours,
Stefano Menini

·

Basic reporting

The work presented by Yuan-yuan Zhou and colleagues shows the performance of a panel of mutations for the stratification of thyroid nodules with indeterminate cytology.

The authors presented a well-written work supported by the literature with a well-explain purpose and good images.

Experimental design

The work presented aligns with the journal's aim and scope.

The author's strategy is well-presented; nevertheless, the paper needs some revisions to be published.
Methods described are sufficiently detailed.

Validity of the findings

The authors presented the mutational analysis performed on nodules from 3 different Bethesda categories (III, IV, V). Moreover, the authors highlight the innovation in presenting the performance data of all three classes using their panel.
Even if certainly not malignant, class V nodules have a higher risk than the other two cytological classes of being so. Given this purpose, it would be appropriate to underline more extensively the differences found among the analyses of the three classes separately and compare the results with those obtained in the comprehensive analysis.

The authors do not report the allele frequencies of the mutations, at least for the cases resulting in false positives or false negatives groups is essential to report the frequency of the mutations found or to check for the presence of low-frequency mutations not highlighted by the automatic analysis software.

Moreover, did the authors request a second pathologist's opinion regarding the false positive and false negative cases? this check should be done.

Regarding the double mutated nodules, many driver mutations of thyroid cancer are usually mutually exclusive. Therefore, validating the presence of the mutations with a second methodology (sanger sequencing; real-time) is mandatory for these cases.

Additional comments

Revision of the nomenclature of mutations is needed, as Shuji Ogino et al. reported in the Journal of Molecular Diagnostics in 2007 (DOI: 10.2353/jmoldx.2007.060081).

Reviewer 2 ·

Basic reporting

The paper is well organized and presented. The aims are clearly presented and follow the aims and scope of the journal. The research question is relevant and interesting to the field
However the Ms has 2 severe drawbacks : the series selection bias and the extremely high prevalence of cancer in the undetermined FNAB series (with a cancer prevalence of 94.71%) that, strongly limits the assumptions and conclusions,

Some overstatements should be corrected e.g. "Innovatively, this study isolated nodules with concurrent molecular alternations from single aberrations and analyzed the two sets of data separately." that approach is not innovative since it was presented before in several papers. Please correct.

Experimental design

The sample collection and selection is not clear, it seems to present a bias. In line 102 the authors state that "From the 529 FNA samples from cytologically indeterminate thyroid nodules (i.e. diagnosed by clinical pathologists as Bethesda III, IV or V)" but in the chart (fig1) they put 832 cases diagnosed as Bethesda I-VI from which 529 are undeterminated. Please clarify.
Even if we consider 832 cases , from which 529 are indeterminate, that gives a very high rate of indeterminate cases (64%) in the series, please explain that issue.
In line 104 the authors state "reports from nodules following definitive surgery (if performed)" . Please indicate clearly, how many cases have definitive histological diagnosis.
In line 167, again the distributions of the Bethesda diagnosis is strange, according to the published prevalence "The majority of thyroid nodules (410 cases, 77.50%) were interpreted as Bethesda V, whereas 14.56% (77 cases) and 7.94% (42 cases) were Bethesda III and IV, respectively." Please comment on that.
The problem of Figure 2 calculations is that the rate of malignancy in the current series is much higher than the proposed values for Bethesda. Please explain that discrepancy.
In lines 205-209 the results raise some concerns. For example in relation to TERT mutations, have the authors check promoter mutations or coding region mutations? The latter are extremely rare in cancer in general. TERT promoter mutations are the ones that have been described in thyroid cancer and the authors do not report any, which is strange. Please clarify that important point. The same for PAX8 that has been involver in thyroid by rearrangements and not point mutations. Have the authors looked for rearrangements?
An additional concern with the Ms is the validation of the genetic alterations.
In particular the " BRAF-PXK fusion, FLNC-BRAF fusion, BRAFN486_T491delinsS, PAX8C147R,
PTEND252Rfs*42 and TERTQ302R have not been reported relative to solid tumors" have been validated by sanger or other method? that must be presented, it could be false /technical issues.
In line 221 the authors refer “in 3 false-negative nodules in Bethesda VI group, and the following 5 cases…” . Have the authors included also Bethesda VI group? That was not stated before….
In line 223 the authors report “1 RETC634R mutants” . The series includes also MTCs? That is not clear in the text. It is also a concern the TP53 mutation "the other one (TP53G262-S269del) turned out to be PTMC on resection though cytologically categorized into Bethesda IV (suspicious for a follicular neoplasia) at the beginning." those mutations are extremely rare in microcarcinomas.
The series bias is also reflected in the prevalence of PTC/PTMC in BRAF-like group reached 100% .
Taking in account that bias, it is more correct if the authors exclud the BRAF-like and RAS-like designations. In fact that designations are based on RNA expression and not in mutation detection (although there is a correlation between them).
Again in lines 275 the authors refer to a Bethesda VI cohort, that do not makes sense and it is not part of the present series. Please delete all the mentions to those cases. "Additionally, similar distribution could also be observed in Bethesda VI cohort, where 108 patients were determined malignant thyroid tumors by both histopathologic and molecular diagnosis and 5 of them were detected concurrent molecular alterations with 8 aberrations involved (data disclosed in Supplemental
Table 2)."
The results section 255-289 is very confuse, based in isolated cases and do not constitutes an organized clinical pathological section. Please reduce to the essential message or delete.
In line 122 " superfluous primers" is repeated.
Line 135 please correct "as well as separately for Bethesda III, IV and IV nodules", must be III, IV and V.
In line 351 please delete "which demonstrated more aggressive features or poorer progression in BRAF-like group compared to those in RAS-like group " since the aggressivness is confered by TP53 and TERT, rather than by BRAF mutation.
Line 150 "Slides of surgical histopathology for special cases were retrospected by another pathologist." Please specify those special cases.
Line 181 "malignant cancer" is redundant. correct please.
Line 202 "which might account for 16 patients of papillary thyroid carcinoma"Why might? no histologic validation on those?
Line 233 Molecular risk stratification (Concurrent molecular alternations excluded). Please correct the typing error.
Line 255 Concurrent molecular alternations. please correct , alterations? Please correct also along the text.
Line 262 and 281- What is NARSQ61R mutant???

In table 2 , which means the Histopathologic diagnosis positive/negativ? is benign/malignant, if that is the case please change . It not please explain.
In Table 3 the Drug resistance column can be deleted since it is out scope and it was not commented in the text. Furthermore, soem of the drugs are not yet approved for the indicated target.
Please exclud table 6 since the information presented is not relevant (many cases TX) and do not add pertinent information to the Ms.

Validity of the findings

The extremely high prevalence of cancer in the series (with a cancer prevalence of 94.71%), strongly limits the assumptions and conclusions, namely in which refers PPV, NPV etc. The authors must stress that limitation in the abstract conclusions, as well as in the paper limitations and in the text conclusions.
In the discussion lines 310-320, the authors must included a caution note , refering that the NPV and PPV assumptions are limited by the extremely high rate of malignancy in the present series. This is mandatory.

Additional comments

In general, the discussion part is extensive and confuse. It deserves a language revision and must be more synthetic.

Reviewer 3 ·

Basic reporting

The English language could be significantly improved to enable clear communication and correct interpretation of the text. Phrases like “applied as a potent auxiliary diagnosis” are confusing.
Line 80 Afirma GEC should not be mentioned as this is an outdated modality no longer available, only GSC should be mentioned for Afirma.
Line 82 Would be helpful for the international audience if the authors could cite a specific self-designed multigene panel on the Chinese market since they mention several verified panels exist.
Line 89 alteration spelled incorrectly. The authors should clarify why they chose to analyze nodules with concurrent molecular alterations separately, what was the hypothesis that drove this decision?
Line 134 typo in repeating Bethesda IV.
Materials and methods section is well written and clear.
Line 143-144 please note the classification of all variants into this 3 category system is provided (e.g. in Supplemental Table 2).
Table 1 do not truncate the p-values, instead use <0.001.
Line 273 “leaf” perhaps “lobe” was intended?
Typo in Line 281, should be “NRAS”

Experimental design

Well designed and executed study, overall.
The concurrent BRAFV600E and RAS mutations should be confirmed both in the FFPE and with an orthogonal modality.
Line 395 mentions longer follow up, this should also be to elucidate the outcomes of unresected indeterminate nodules that prospectively underwent analysis of the panel in this study.

Validity of the findings

With respect to the BRAFV600E concurrent with RAS mutations, the authors correctly mention that these alterations have been shown to be mutually exclusive. However, Lines 353-359 attempt to explain this but are not coherent. This is a surprising finding and important if true, and should be explained clearly in much more detail. Could there be two adjacent nodules sampled with different driver mutations instead of truly concurrent mutations within a homogeneous nodule in these cases?
90.91% prevalence of malignancy in Bethesda III is way out of the range for Bethesda III in the guidelines or the literature. This needs to be addressed in more detail, and is a serious detriment to generalizability of the results. Even 66.67% prevalence of malignancy in Bethesda IV is higher than expected, and because most nodules in this study are Bethesda V which is known to have very high prevalence of malignancy, the NGS test used in this study has functionally not been validated in benign thyroid nodules (n=28 histologically benign nodules). This is a significant limitation and unfortunately considerably reduces the value of the results, although of course the number of surgeries cannot be controlled with the study design, the authors should show whether this extremely high malignancy rate is consistent with long-term malignancy rates in each Bethesda category in their institution or is skewed by limited follow up or other factors.
Line 397 Seems odd to question the malignant potential of nodular goiter, where was this described in the results?
Line 398 The authors should include additional limitations, namely the high prevalence of malignancy in the indeterminate categories and the low sample size of Bethesda III and IV nodules.

Additional comments

I commend the authors for the tables and figures, particularly for sharing the raw data in Supplementary Table 2 which is a welcome addition to the literature that often provides less comprehensive patient level data.

---

## Round 0.2 · Major Revisions

Dear Dr. Ding,

Your manuscript entitled “Performance of a multigene testing in cytologically indeterminate thyroid nodules and molecular risk stratification"" has again been carefully reviewed by the Editor and Reviewers.

Two Reviewers still raise several issues concerning the experimental design, the validity of the findings, wording and structure, and inaccurate or incomplete description that need to be addressed.

If you decide to resubmit a re-revised version of your manuscript, please summarize all the improvements made in the new version and give answers to all critical points raised in the reviewers’ report in an accompanying letter. Please copy and paste each and every reviewer's comment above your response. If you feel any of their points are inappropriate, you are certainly free to provide rebuttal in your covering letter.

Please note that resubmitting your manuscript does not guarantee eventual acceptance. I reiterate that the acceptability of the revision will depend upon the resolution of all the points raised by the reviewers.

Sincerely yours,
Stefano Menini

Reviewer 2 ·

Basic reporting

The revised version of the Ms, has been partly modified, and the English was partly reviewed, although some unclear sentences remain.
Example
Lines 1091 to 1094 – “Among 174 BRAF-like tumors with confirmed subtyping, 57 cases (32.76%) were observed infiltration of thyroid capsule (T4 staging) and 115 cases (45.28%) of 254 assessable data happened lymph node metastases (N1 staging). In RAS-like groups, however,” , please correct the english, here and through the Ms.

Experimental design

Line 1411 – “Additionally, the extremely high prevalence of cancer in the cohort should be noted for clinical reference.” Please elaborate more on that, what do the authors mean with "clinical reference "?
In line 1739 the authors refer that “The BRAFV600E and mutated RAS were considered to be two mutually exclusive drivers of PTC, possibly suggesting similar or redundant downstream effects, and causing different signaling effects followed by profound phenotypic differences (Ren et al., 2022)” however they do not elaborate on the fact that their results report a significant number of cases with concomitant BRAF/RAS mutation. Please discuss that point. The reason presented in line 1756 “These ambiguous discrepancies may have resulted from an optimizable computerized algorithm,” is not clear.
My concern in relation to TERT mutations is maintained. Have the authors checked promoter mutations or coding region mutations? The latter are extremely rare in cancer in general. TERT promoter mutations are the ones that have been described in thyroid cancer and the authors do not report any, which is very strange. Please clarify that important point. The same for PAX8 that has been involved in thyroid by rearrangements and not point mutations. Have the authors looked for rearrangements?
The concern with the Ms relating with the validation of the genetic alterations. In particular the " BRAF-PXK fusion, FLNC-BRAF fusion, BRAFN486_T491delinsS, PAX8C147R, PTEND252Rfs*42 and TERTQ302R that have not been reported relative to solid tumors, was not resolved. That must be presented; it could be false /technical issues.
The title of Table 3 should be modified. It refers only to a subset of cases of the study, with rare mutations
Although the authors have taken out the MTC case in the text, it is still present in the Table 5 , please delete.

Validity of the findings

The major limitation of the Ms is maintained, and the justification for that bias is not totally clear.The extremely high prevalence of cancer in the series (with a cancer prevalence of 94.71%), strongly limits the assumptions and conclusions, namely in which refers PPV, NPV etc.

Reviewer 3 ·

Basic reporting

Previous comments have been addressed.

Experimental design

Previous comments have been addressed.

Validity of the findings

I understand you may not have consent or access to test tissue and am glad to hear a prospective study is being conducted, but in the meantime without orthogonal testing the concurrent mutually exclusive mutations should be considered putative at best. This finding goes against genomic data from thousands of indeterminate nodules, and has not been adequately validated in your study. Please suggest other possible explanations for this finding, such as contamination or sampling of adjacent nodules.

Additional comments

The extremely high malignancy rate in indeterminates in this study remains problematic, but has been sufficiently pointed out by the authors in the text as a limitation. It would be helpful to add one or two sentences in the manuscript explaining possible reasons for the unusually high malignancy rate.

---

## Round 0.3 · accepted · Accept

Dear Dr. Zhou,

Thank you for submitting a revised version of your manuscript. I am pleased to inform you that your manuscript is accepted for publication in PeerJ. Some corrections to the text are needed and can be done during proofreading.

I thank all reviewers for their effort in improving the manuscript and the authors for their cooperation throughout the review process.

Sincerely yours,
Stefano Menini